# Right Ventricle and Radiotherapy: More Questions than Answers

**DOI:** 10.3390/diagnostics13010164

**Published:** 2023-01-03

**Authors:** Marijana Tadic, Johannes Kersten, Dominik Buckert, Wolfgang Rottbauer, Cesare Cuspidi

**Affiliations:** 1Klinik für Innere Medizin II, Universitätsklinikum Ulm, Albert-Einstein Allee 23, 89081 Ulm, Germany; 2Department of Medicine and Surgery, University of Milano-Bicocca, 20126 Milano, Italy

**Keywords:** right ventricle, radiotherapy, cancer, systolic function

## Abstract

The injury of the left ventricle (LV) during anticancer therapy has long been recognized, and guidelines recommend a specific set of parameters for determination of LV impairment. The influence of anticancer therapy on the right ventricle (RV) has been insufficiently investigated, and there are only a few studies that have considered the effect of radiotherapy on RV remodeling. On the other hand, large number of patients with different types of cancers located in the chest are treated with radiotherapy, and the negative clinical effects of this treatment such as accelerated coronary artery disease, valve degeneration and heart failure have been documented. The anatomical position of the RV, which is in the front of the chest, is responsible for its large exposure during radiation treatment, particularly in patients with left-sided breast and lung cancers and mediastinal cancers (hematological malignancies, esophagus cancers, thymomas, etc.). For the same reason, but also due to its anatomical complexity, the RV remains under-investigated during echocardiographic examination, which remains the cornerstone of cardiac imaging in everyday practice. In the last decade many new echocardiographic imaging techniques that enable better evaluation of RV structure, function and mechanics appeared, and they have been used in detection of early and late signs of RV injuries in oncological patients. These investigations are related to some important restrictions that include limited numbers of patients, used parameters and imaging techniques. Many questions about the potential impact of these changes and possible predictions of adverse events remain to be evaluated in future large longitudinal studies. The current body of evidence indicates an important role of radiotherapy in RV remodeling, and therefore, the aim of this review is to summarize currently available data regarding RV changes in patients with various oncological conditions and help clinicians in the assessment of possible cardiac damage.

## 1. Introduction

Radiotherapy is a very important part of anticancer therapy in a large number of oncological patients. However, it may be related with cardiac damage in cancers with chest localization—such as breast, lung, esophageal and mediastinal cancers including lymphomas—as well as some pediatric cancers such as lymphoma, thymoma and teratoma. The influence of radiotherapy is difficult to differentiate from chemotherapy-induced effects, as they are commonly used together and their individual effects are difficult to evaluate.

Chemotherapy is known to induce early cardiotoxicity, whereas radiotherapy has been considered to be more related to late cardiac damage. However, the use of novel imaging methods, primarily in echocardiography, has enabled the early detection of cardiac impairment in oncological patients treated with radiotherapy [1]. Novel cardiac-sparing techniques used in radiotherapy have significantly reduced the level of radiation exposure of all tissues, and particularly of those tissues that are not related to the cancer. Nevertheless, various parts of the heart are radiated and affected differently, which mainly depends on tumor localization [1]. The majority of studies have investigated the effect of radiotherapy on the left ventricle (LV) [2,3], whereas the effect on the right ventricle (RV) has been studied significantly less [4,5,6]. Novel reports are more focused on subclinical LV changes that correspond with the deterioration of LV multidirectional strain [2,3]. The predictive value of LV functional changes, primarily a reduction in systolic function measured by LVEF, is well-established and recognized by all guidelines, while the predictive function of RV remodeling has recently been documented in cancer patients treated with chemotherapy [7,8]. The anterior anatomic position of the RV in the chest makes this cardiac chamber very exposed to radiation in a large number of left-sided chest cancers, as well as mediastinal tumors. It is difficult to distinguish effect of radiotherapy on the LV from that on the RV because there is significant interventricular dependence, and additional effects of thickening and stiffening of the pericardium may increase RV and LV end-diastolic pressures. Different imaging parameters are used to overcome these issues with more or less success.

The PubMed database was searched for all articles published in English up to 31 October 2022. Studies were identified by using the following terms: “right ventricle”, “cancer”, “radiotherapy”, “strain”, “systolic function”, “longitudinal strain” and “echocardiography”. The number of studies investigating the relationship between radiotherapy and RV remodeling is scarce, and the number of included patients is very limited. However, the majority of these studies indicate significant changes in RV systolic function and mechanics not only during long-term follow-up, but also in a period of few weeks and months after exposure to this type of therapy. The aim of this article is to summarize data about RV structural, functional and mechanical remodeling in patients with different types of cancer in order to help clinicians in treating and following up with these patients.

## 2. Radiotherapy and Cardiac Damage—Pathophysiology

The cornerstone responsible for almost all cardiac changes found in radiation-induced heart disease is tissue fibrosis that is related to micro- and macrovascular injury. Microvascular injury is related to myocardial fibrosis, pericardial fibrosis and decreased capillary density. These changes induce LV remodeling and changes in diastolic and systolic function, as well as pericardial effusion and constriction. Macrovascular injury is associated more with accelerated coronary atherosclerosis, which induces acute and chronic myocardial ischemia.

Cellular damage induces vasodilation and vascular permeability even after a few minutes of ionizing radiation. Impaired endothelial cells release adhesion molecules and growth factors responsible for the acute inflammatory response that includes the secretion of inflammatory cytokines (monocyte chemotactic factor, tumor necrosis factor (TNF) and interleukins (IL) including IL-1, IL-6 and IL-8) [9]. The most important factors in the acute phase are neutrophils, which may be found in all cardiac layers in regions exposed to radiotherapy. Matrix metalloproteinases damage the endothelial basement membrane, which activates pro-inflammatory cells in the locations of damaged tissue in order to heal it. The initial microvascular damage also initiates the coagulation cascade, which results in rapid fibrin deposition. The acute phase occurs only several days after radiotherapy. The chronic inflammation includes the release of profibrotic cytokines such as platelet-derived growth factor, transforming growth factor β, basic fibroblast growth factor, insulin-like growth factor and connective tissue growth factor. This leads to fibrin deposition, fibrosis, scarring and ultimately impaired cardiac function.

The classical hallmarks of radiation-induced heart disease (RIHD) include the following: fibrosis and calcification of the aortic root and the aortomitral curtain that can lead to progressive stenosis of the aortic and mitral valves; ostial coronary stenosis; myocardial atrophy; and widespread pericardial adhesions and thickening ultimately leading to intractable and inoperable pericardial constriction. However, one should be aware of the possibility that the presence of cancer itself has a potentially negative influence on RV functioning and mechanics, as we reported in a cohort of patients with different types of solid cancers [10].

Interestingly, similar mechanisms responsible for RIHD might induce changes in cardiac metabolism, myofibril disorganization, imbalance among protein synthesis and degradation, cellular apoptosis and ultrastructural changes of cardiomyocytes that altogether may trigger cancer-induced cardiac dysfunction. The interaction between inflammation, oxidative stress, sympathetic system hyperactivity and vasoactive cytokines that induces myocyte damage may be related with cancer-induced RV impairment.

## 3. Breast Cancer

Breast cancer is the most common cancer in women, and improved diagnostics and adjuvant therapies have significantly improved survival rates in these patients. Adjuvant radiotherapy has an important role in anticancer therapy of these patients. However, cardiac exposure is known to be associated with adverse cardiovascular effects including impairment in cardiac function and mechanics, but also valvular damage as well as advanced coronary artery disease. Therefore, it is very important to provide short- and long-term follow-up for these patients in order to provide timely diagnosis and treatment of radiation-induced heart disease. There is an obvious difference in cancer location, and the cardiac exposure is essentially different between left- and right-sided breast cancer.

A study that investigated 49 patients with left-sided breast cancer did not find significant differences in conventional parameters of RV systolic and diastolic function before and approximately 40 days after radiotherapy, with the exception of tricuspid annular plane systolic excursion—TAPSE—which was significantly reduced after radiotherapy was applied [11]. The same group of authors investigated whether adding aromatase inhibitors to radiotherapy in breast cancer patients influenced RV function and reported more pronounced deterioration of TAPSE in patients with left-sided breast cancer treated with both treatment modalities than in those women treated only with radiotherapy [12]. Nevertheless, other parameters of RV systolic and diastolic function remained the same after anticancer therapy. Table 1 summarizes available data regarding the effect of radiotherapy on RV remodeling in patients with breast cancer.

The same research group provided a 3-year study of 80 patients with left- and right-sided breast cancer treated with radiotherapy and reported a significant short-term reduction in TAPSE, but not in other parameters of RV systolic function (s’) or diastolic function (E/e’) [4]. The majority of patients (*n* = 60) who experienced short- and long-term TAPSE reduction had left-sided breast cancer. RV systolic function deterioration was not found in patients with right-sided breast cancer (*n* = 20) during short- and long-term follow-up. Investigators additionally studied ultrasound tissue characterization 34 ± 11 days after radiotherapy in these patients and found elevated echodensity in the RV free wall and in the septum in patients with left-sided breast cancer, whereas there were no changes in patients with right-sided breast cancer [13]. RV and septal echodensity were increased in >60% of patients with left-sided breast cancer, and this condition was inversely associated with TAPSE, which was proven to be a reliable and reproducible RV systolic function parameter and early indicator of RT-induced RV changes [13].

A small study that used cardiac magnetic resonance and followed 20 patients with breast cancer for 12 months showed impairment in RV mechanics that started early after radiotherapy application [14]. Namely, after 6 weeks there was already a trend towards reduced global, mid and apical RV longitudinal strain (*p* = 0.06), whereas after 12 months, global longitudinal RV function was significantly reduced (−13.9% vs. −15.1%, *p* < 0.05). The reduction in RV longitudinal strain was mostly caused by a decrease in apical strain, which is concomitantly the field that receives the highest radiation dose. Nevertheless, different receptors are present in the three myocardial layers, as well as in various cardiac segments, which might also be the reason for the unique response of each myocardial region (basal, mid and apical).

RV diastolic parameters were not deteriorated in women with breast cancer treated with radiotherapy compared to controls, whereas RV structure was mostly not investigated [4,11,12,13]. The obvious limitation of these studies is that majority of them (four out of five) come from the same group of authors and the same center. Limited numbers of patients and short follow-up periods are additional important weaknesses of these investigations. RV mechanics were evaluated only in the study that used cardiac magnetic resonance, not echocardiography, which represents the most widely and frequently used imaging technique for cardiac assessment.

## 4. Hematological Malignancies

The largest echocardiographic investigations in this field studied surviving patients with hematological malignancies that were treated during childhood. They provided comprehensive data regarding RV remodeling in both genders and very long follow-up periods (>10 years), which is very different from studies performed in patients with breast cancer that included only women and had significantly shorter monitoring.

A study included 246 survivors of childhood lymphoma and acute lymphoblastic leukemia, mean 21.7 years after diagnosis, and 211 matched controls [6]. The authors reported that all parameters of RV function and mechanics were lower in the survivor group than in controls (fractional area change (FAC), TAPSE, peak systolic tricuspid annular velocity (s’) and free-wall strain). However, the majority of patients were treated with anthracycline therapy alone (*n* = 149) or in combination with radiotherapy (*n* = 40), whereas only small portion of patients (*n* = 18) was treated with only mediastinal radiotherapy. Even though all three groups had significantly lower values of RV systolic function (TAPSE, FAC and s’) and RV free-wall strain, patients treated only with mediastinal radiotherapy had a trend of lower values than other patients [6]. Statistical significance was not reached due to the small sample size, but the interesting trend toward worse parameters of RV systolic function in patients treated only with radiotherapy was also acknowledged by the authors. The study showed a significant percentage (~40%) of patients with overlapping left and right ventricular failure, whereas isolated RV systolic dysfunction was found in about 10% of all cancer survivors.

Another large study involved 274 lymphoma survivors who were followed for 13 ± 6 years and separated patients into those who were treated only with anthracyclines and those who were treated with anthracyclines plus radiotherapy, as well as 222 control subjects [15]. Moreover, the investigators divided both groups into subgroups according to the level of received anthracycline therapy (cut-off: 300 mg/m^2^) and radiation (cut-off: 30 Gy). The patients who received both anthracycline therapy and radiation >30 Gy were found to have the worst RV systolic function (TAPSE, FAC and s’) and RV longitudinal mechanics (global and free-wall strain) [15]. However, the difference did not reach statistical significance. In a multivariable analysis, all parameters of RV systolic function were impaired in patients treated with high-dose radiotherapy compared with patients who received only chemotherapy. The subgroup that received low-dose radiotherapy exhibited impaired RV FAC and greater RV strain in comparison to those who did not receive radiotherapy, whereas the other parameters were similar among these groups. Interestingly, the authors did not find significant differences in the parameters of RV systolic function between patients treated with low-dose versus higher-dose anthracycline therapy without concomitant radiotherapy. They also did not report an association between the cumulative dose of anthracyclines and RV systolic functional parameters [15].

The authors also investigated the relationship between RV remodeling and functional capacity, evaluated by a cardiopulmonary exercise test (peak oxygen consumption—peak VO2) [15]. TAPSE, RV free-wall longitudinal strain and RV index of myocardial performance correlated with peak VO2, whereas FAC and s’ did not show such a correlation. Nevertheless, after LV longitudinal strain was included in the model, the associations between TAPSE, RV free-wall longitudinal strain and RV index of myocardial performance and peak VO2 vanished.

RV thickness was 14% lower in patients compared with control subjects. Patients had significantly smaller RV mid and basal diameters and RV end-diastolic areas. Considering the fact that the numerical values were small, the authors concluded that there were no clinically important differences in RV morphology between the observed groups [15]. RV diastolic function was not evaluated in either of these two studies, and it is not possible to speculate about it in the patients with hematological malignancies. The findings from these large studies are summarized in Table 2.

There are some evident limitations of these investigations. Namely, a long-term follow-up without echocardiographic assessment in the meantime cannot provide evidence that radiotherapy and/or chemotherapy were the only causes of RV remodeling, as in the meantime these patients developed comorbidities that also might induce changes in RV structure, function and mechanics. Even though the majority of well-known comorbidities (hypertension, diabetes and obesity) were not very prevalent in these patients’ populations and were included in the multivariable analysis, it is difficult to completely exclude their influence. The patients were not regularly followed, and therefore it is not clear when RV remodeling started. It is possible that it began much earlier, but it can be also hypothesized that RV deterioration was gradual and progressive.

## 5. Thoracic Cancers (Lung and Esophagus)

There is a substantial lack of data about the effect of radiotherapy on RV remodeling in patients with other types of cancers. A study that included 70 patients with esophageal cancer, central lung cancer, thymoma and left breast cancer showed that TAPSE and the Tei index were sensitive indicators for the early detection of right heart injury after RT for thoracic tumors [5]. Nevertheless, the 3D RV ejection fraction revealed no significant change in the early stage of right heart damage after radiotherapy. In acute radiation-induced right heart injury, the combined application of the RV Tei index, tricuspid annular displacement and NT-proBNP is clinically relevant.

A small study that investigated the influence of low-dose radiotherapy in patients with lung and esophageal cancer found a significant reduction in TAPSE [16]. The limitation of these studies are small sample size, different cancer localizations and very short follow-up periods (4–6 weeks). Additionally, results about changes in RV structure, diastolic function and mechanics are still not available.

Chen et al. investigated 128 inoperable patients with stage III non-small-cell lung cancer (NSCLC) before and after concurrent chemoradiotherapy that included platinum agents (57.5% platinum/Taxol doublet, 42.5% platinum/non-Taxol doublet) and radiotherapy with a median dose of 64.5 Gy that was administered over 5–7 weeks [17]. The comprehensive assessment of RV structure, function and mechanics was performed at baseline and 6 months after therapy was finished. The findings did not show any difference in RV structure (diameters) and systolic function (TAPSE, FAC and s’), but there was a significant deterioration in RV global and free-wall longitudinal strain [17]. Interestingly, a modest correlation was found between the changes in RV strain and radiation doses to the corresponding cardiac structures. The change in RV global and free-wall longitudinal strain significantly correlated with the heart mean dose, the RV mean dose and the RV free-wall mean dose. This emphasized the importance of RV strain evaluation, as it appears to be a more sensitive parameter of RV injury than conventional parameters and correlated well with amount of radiation that the RV received. One of the most relevant findings of this study was the independent predictive value of RV free-wall longitudinal strain at baseline, as well as its change after anticancer therapy, for 6-month all-cause mortality of stage III NSCLC patients [17]. The authors determined that the cut-off for clinically significant reduction in RV free-wall longitudinal strain in their population of patients was 10.1% in absolute values, which is the first suggestion of this kind in the literature. Table 3 provides data about RV remodeling in patients with different types of cancers located in the thorax and mediastinum who were treated with radiotherapy.

## 6. Clinical Implications

Radiotherapy remains an important part of anticancer therapy in various types of cancers, and its impact on cardiac structure, function and mechanics is very important, as these parameters showed significant sensitivity in recognizing subclinical cardiac damage and excellent prediction of cardiovascular and overall morbidity and mortality. The available studies, although not sufficient, provide evidence about RV functional changes in radiotherapy-treated cancer patients, which should be enough to justify the inclusion of RV parameters in routine echocardiographic examinations (basic and follow-ups) in all cancer patients. Follow-up should start shortly after radiotherapy, but is has to be prolonged for decades after therapy is completed. A limited number of studies revealed the large significance and clinical importance of RV longitudinal strain determination, which seems to be more sensitive in detection of RV injury than conventional echocardiographic parameters; it is also associated proportionally with the amount of radiation that RV myocardium received and is a good predictor of all-cause mortality [17,18]. The present review summarizes currently available data about RV remodeling in patients with different types of cancer who are most frequently treated with radiotherapy, and clinicians treating various groups of patients can easily find information related to their field of interest, which represents a certain clinical implication of the current article. Table 4 provides an overview of all RV parameters and their changes in oncological patients after radiotherapy. It is based on the main findings from major studies because there is lack of agreement about some parameters.

The role of RV longitudinal strain should be particularly emphasized in the early detection of RIHD. The effect of radiotherapy has been more extensively investigated regarding LV multidirectional strain, particularly global longitudinal strain (GLS). However, available studies show that RV GLS is a more reproducible and sensitive method for the detection of subclinical cardiac impairment than TAPSE, FAC and s’ [18]. Nevertheless, in these studies chemotherapy and radiotherapy were used together, and it is difficult to distinguish the effect of chemotherapy from that of radiotherapy. Other studies, not related with cancer, proved that RV GLS is good predictor of adverse outcomes in patients with various cardiovascular conditions [19,20]. Therefore, it is recommended to assess RV GLS at the baseline examination, as well as during each follow-up visit.

The importance of CMR in the detection of RV impairment in cancer patients has been poorly investigated, and it is difficult to provide recommendations based on the available data. However, if we examine the existing guidelines related to multimodality imaging in patients treated with radiotherapy for the LV, we may recommend CMR in situations when echocardiographic images are of limited quality or when additional information is required (e.g., suspicion of myocarditis or ischemic disease). Routine CMR for evaluation of the RV in oncological patients is not recommended due to its low availability, its costs and the limited number of healthcare professionals with the required level of expertise for this kind of examination.

The duration of follow-up in oncological patients treated with radiotherapy should last as long as possible, at least 5–20 years after radiation exposure, but preferably decades after radiation because studies showed that deterioration in LV and RV function remains for more than two decades after therapy, perhaps even longer; however, studies with longer follow-up periods are not available [21]. Moreover, heart valve impairment occurs 10–15 years after radiotherapy. The frequency of follow-up examinations depends on the level of radiation and concomitant chemotherapy received, as well as RV damage found during the last examination. In patients without pre-existing cardiac abnormalities, annual echocardiographic examination would be recommended during the first 10 years after treatment and afterwards every 5 years. In patients with pre-existing cardiac abnormalities or damage that was found during follow-up, surveillance echocardiographic examinations should be performed more frequently depending on the diagnosis (deterioration of cardiac function, valve disease or coronary artery disease), and additional techniques might be also introduced (stress echocardiography and stress CMR).

## 7. Conclusions

Currently available data show that the RV cannot be considered an insignificant cardiac chamber and has to be involved in the routine echocardiographic evaluation of patients with cancer treated with chemotherapy or radiotherapy. The effect of cancer itself should be also carefully considered at the baseline examination, before anticancer therapy is initiated, as some evidence about RV remodeling in untreated patients shows that chemo- and radiotherapy are not solely responsible for RV injury. The type of treatment (chemotherapy with or without radiotherapy or isolated radiotherapy) is also very important for RV remodeling because the cumulative dosage of medications and radiation has a determinant role in RV damage. An increasing body of evidence implies that RV involvement in cancer-related cardiotoxicity is very frequent, but its impact on outcomes should be established in large prospective investigations. These studies should also provide information about whether some cardioprotective therapy should be used in patients undergoing radiotherapy with or without chemotherapy in order to prevent potential cardiac remodeling.

## Figures and Tables

**Table 1 diagnostics-13-00164-t001:** RV remodeling in patients with breast cancer after radiotherapy.

Reference	Left-Sided Cancer	Right-Sided Cancer	Follow-up Period	Main Findings
Skyttä et al. [4]	60	20	3 years	The reduction of TAPSE in left-sided but not right-sided breast cancer. The increase in the tricuspid insufficiency gradient in both groups. RV diastolic function (E/e’) and tissue-Doppler parameter of RV systolic function (s’) did not change in any of these groups over 3 years.
Tuohinen et al. [11]	49	/	41 ± 11 days	TAPSE was the only RV parameter that was impaired after radiotherapy. RV diameter, s’ and E/e’ (parameters of RV structure and systolic and diastolic function) did not change during this short follow-up.
Skyttä et al. [12]	60	/	60 days	TAPSE was reduced in patients in both groups, treated with radiotherapy with or without aromatase inhibitors, but the decline was higher in those who were treated with both therapies. E/e’, s’ and the tricuspid insufficiency gradient did not change during the course of therapy in any of these groups.
Tuohinen et al. [13]	58	20	39 ± 13 days	Elevated echodensity of the RV free wall in patients with left-sided breast cancer and no changes in patients with right-sided breast cancer. Reduced TAPSE only in patients with left-sided cancer. E/e’, s’ and the tricuspid insufficiency gradient remained unchanged after therapy.
Femia et al. [14]	20 *	/	12 months	After 6 weeks of radiotherapy there was a trend towards reduced global, mid and apical RV longitudinal strain (*p* = 0.06), whereas after 12 months RV GLS showed statistically significant reduction.

* not specified on which side the breast cancer is, GLS—global longitudinal strain, RV—right ventricle, s’—systolic velocity of the lateral segment of tricuspid annulus, TAPSE—tricuspid annular plane systolic excursion, TI—tricuspid insufficiency.

**Table 2 diagnostics-13-00164-t002:** RV remodeling in survivors with hematological malignancies who received radiotherapy.

Reference	Lymphoma	Leukemia	Follow-up Period	Main Findings
Christiansen et al. [6]	115	131	21.7 years	All parameters of RV systolic function (TAPSE, FAC and s’) and RV free-wall longitudinal strain were significantly lower in patients treated with mediastinal radiotherapy with or without anthracyclines. All parameters were the worse in patients treated only with mediastinal radiotherapy compared with patients treated with chemotherapy, even when combined with radiotherapy. Furthermore, 50.5% of patients treated only with mediastinal radiotherapy had impaired RV function in comparison to those treated with anthracyclines (28.2%) or anthracyclines and radiotherapy (32.5%).
Murbraech et al. [15]	274	/	13 ± 6 years	Patients treated with radiotherapy had significantly lower RV thickness compared with controls. There was no difference in RV diameters between survivors treated with radiotherapy and those treated with chemotherapy. All parameters of RV systolic function (TAPSE, s’ and FAC) and RV mechanics (RV GLS and free-wall longitudinal strain) were significantly lower in patients treated with radiotherapy +/− anthracycline therapy. Greater cardiotoxic burden was related with larger RV functional damage. TAPSE correlated with functional capacity (peak oxygen consumption) in all survivors.

FAC—fractional area change, GLS—global longitudinal strain, RV—right ventricle, s’—systolic velocity of the lateral segment of tricuspid annulus, TAPSE—tricuspid annular plane systolic excursion.

**Table 3 diagnostics-13-00164-t003:** RV remodeling in patients with thoracic cancer who received radiotherapy.

Reference	Number of Patients	Type of Cancer	Follow-up Period	Main Findings
Li et al. [5]	70	Esophageal, central lung, thymoma, and left breast	4 weeks	The Tei index significantly increased in the second week of radiotherapy, whereas TAPSE decreased significantly 4 weeks after radiotherapy. 3D RVEF did not change after radiotherapy.
Slager et al. [16]	29	Esophageal and lung	6 weeks	TAPSE reduced proportionally to radiation dose exposure.
Chen et al. [17]	128	Non-small-cell lung cancer (stage III)	6 months	Global and free-wall RV longitudinal strain reduced 6 months after radiotherapy. RV radiation mean dose correlated with reduction of RV free-wall longitudinal strain, which is an independent predictor of all-cause mortality.

RV—right ventricle, RVEF—right ventricular ejection fraction, s’—systolic velocity of the lateral segment of tricuspid annulus, TAPSE—tricuspid annular plane systolic excursion.

**Table 4 diagnostics-13-00164-t004:** Echocardiographic parameters used for evaluation of RV function.

Parameter	Echocardiographic Window	Changes	Limitations
**Parameters of RV structure**			
RV diameters	4-Chamber view	→	Depending on RV shape, possible underestimation of RV dilatation
RA area and volume	4-Chamber view	→	Biplane not feasible in majority of patients and therefore frequently inaccurate Lack of clear reference values
RV thickness	Subcostal view	↓	Highly dependent on image quality Sometimes difficult to distinguish from pericardial fat and fibrosis
**RV systolic function**			
**2D echocardiography**			
TAPSE	4-Chamber view	↓	Angle-dependent Shows only longitudinal function
FAC	4-Chamber view	↓	Neglects the effect of RVOT on systolic function Modest reproducibility
s’	4-Chamber view	↓	Angle-dependent Shows only longitudinal function
RIMP (Tei index)	4-Chamber view	↑	Unreliable when RA pressures are increased
**3D echocardiography**			
RVEF	Acquisition in modified 4-Chamber view (focused on the RV)	↓	Highly dependent on image quality Load-dependent Significantly lower availability of 3DE than 2DE Reference values not fully established
**RV diastolic function**			
E/A	4-Chamber view	→	Modest reproducibility Reference values not fully established
E/e’	4-Chamber view	→	Modest reproducibility Angle-dependent
**RV mechanics**			
**Global longitudinal strain**	Modified 4-Chamber (focused on the RV)	↓	Vendor dependent Influence of LV (inteventricular septum)
**RV free-wall longitudinal strain**	Modified 4-Chamber (focused on the RV)	↓	Highly dependent on image quality Vendor dependent

E/A—ratio between tricuspid early and late flow velocity measured by pulsed Doppler, E/e’—ratio between early tricuspid velocities measured by pulsed and tissue Doppler, FAC—fractional area change, RIMP—right ventricular myocardial performance index, RV—right ventricle, RVEF—right ventricular ejection fraction, s’—systolic velocity of the lateral segment of tricuspid annulus, TAPSE—tricuspid annular plane systolic excursion. → without change, ↓ decreased, ↑ increased.

## Data Availability

Not applicable.

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
