# Peer review of "Right Ventricle and Radiotherapy: More Questions than Answers"

_diagnostics, 2023, doi:10.3390/diagnostics13010164_

Round 1

Reviewer 1 Report

The authors provided an interesting overview about the influence of radiotherapy on RV remodeling in patients with cancer. Considering the large number of cancer patients treated with this type of therapy and small number of studies on topic, I believe that this manuscript add significant amount of information on this topic. I would have only a few minor suggestions for the authors to improve the reading of their article:

1. RV strain is becoming more important in clinical practice and even though not sufficient data are available, I would ask authors to emphasize the importance of RV strain in daily practice in cancer patients treated with radiotherapy.  
2. What authors would suggest for usage of CMR in patients with cancer? As experts in field , do they have opinion when CMR should be involved in diagnostic of RV remodeling in these patients?
3. In clinical implications authors summarized current practice. Is there some particular groups of patients in which we should be more careful during short and long follow-up?
4. How long you propose the duration of follow-up and how frequently echocardiographic examinations should be performed?
5. There are some small spelling mistakes. Please correct them. 

Author Response

Thank you very much for your observations and time that you put into evaluation of our review article. We have accepted your suggestions and revised the article according to them.

  1. RV strain is becoming more important in clinical practice and even though not sufficient data are available, I would ask authors to emphasize the importance of RV strain in daily practice in cancer patients treated with radiotherapy.  

In the segment about clinical implications we added the paragraph about the usefulness of RV strain in these patients.

2. What authors would suggest for usage of CMR in patients with cancer? As experts in field , do they have opinion when CMR should be involved in diagnostic of RV remodeling in these patients?

We agree with you, CMR is indeed an important imaging modality, but there are many limitations for its usage in daily practice. We provided our opinion based on our knowledge , existing evidence and recommendation about evaluation of LV damage. 

3. In clinical implications authors summarized current practice. Is there some particular groups of patients in which we should be more careful during short and long follow-up?

Thank you for this comment. We mentioned this issue in the paragraph about clinical implications.

4. How long you propose the duration of follow-up and how frequently echocardiographic examinations should be performed?

This topic has been also covered in the clinical implications section.

5. There are some small spelling mistakes. Please correct them. 

We corrected all mistakes that were found in the manuscript and we hope that English is much better in the current form of the manuscript.

We did our best to change manuscript according to your suggestions and provide all necessary explanations. We hope that the manuscript in its current form fulfills your and the Journal criteria for publication.

Reviewer 2 Report

The manuscript entitled: ``Right ventricle and radiotherapy: More questions than answers`` summarized currently available data regarding RV changes in patients with breast cancer, hematological malignancies and thoracic cancers.

It is very well structured and written. This narrative  review is very important in clinical practice.

It is obviously that RV was less studied in cancer patients, especially after radiotherapy.

However, I recommend introducing in chapter entitled clinical implication a table with imaging parameters (frequently echocardiographic parameters) which was used until now in RV assessment in patients with cancer, highlighting their changes.

In addition, there are some minor wording errors.

Page 4 line 151: ``…there were no changes in in patients with right-sided breast cancer.``

Page 5 line 186: ``….mechanics were lower in the survivor group than in controls. fractional area change,..``

Page 6 Table 2: ``…were significantly lower in survovors treated with…``

Author Response

Thank you very much for your comments. We accepted all of them and provided additional table (Table 4) summarizing data from the major studies about different parameters of RV structure, systolic and diastolic function, as well as mechanics. Mistakes in English were also corrected.

We did our best to change manuscript according to your suggestions and provide all necessary explanations. We hope that the manuscript in its current form fulfills your and the Journal criteria for publication.

Reviewer 3 Report

This is the review of radiotherapy and right ventricular function, and I think it is well organized.

#1 The title is in relation to radiotherapy, although many of the reports seem to be in conjunction with chemotherapy. The authors should report if you can assess how much of an effect radiotherapy alone had.

#2 There seems to be a report of a decline in right heart function that persisted for 3 years. Are the effects of radiotherapy permanent, or is there a recovery? The authors had better comment on this. 

#3 In some breast cancers, the effects of radiation therapy have been reported to be different between right breast cancer and left breast cancer. Is this true in other reports as well?

#4 What measures, if any, are being taken to avoid the effects of radiotherapy on cardiac function (including the right heart)?

Author Response

We would like to thank you for the detailed review of our manuscript. We greatly appreciate the effort you made concerning your critique for the review of our study.

#1 The title is in relation to radiotherapy, although many of the reports seem to be in conjunction with chemotherapy. The authors should report if you can assess how much of an effect radiotherapy alone had.

It is very difficult to distinguish effect of radiotherapy from chemotherapy as the majority of patients were treated with both types of anti-cancer therapy. However, we presented also studies in which patients were treated only with chemotherapy and combination therapy and those treated with combined therapy are suffering significantly more cardiac damage than those treated only with chemotherapy. 

#2 There seems to be a report of a decline in right heart function that persisted for 3 years. Are the effects of radiotherapy permanent, or is there a recovery? The authors had better comment on this. 

Actually study showed permanent reduction in TAPSE, whereas s' improved. Strain was unfortunately not determined. Studies with hematological malignancies proved that RV systolic function and GLS remained reduced even 20 years after therapy. We believe that reduction in RV systolic function and mechanics remained permanently. However,  these parameters mostly remain in the range of normal values, even though they are lower than in control group.

#3 In some breast cancers, the effects of radiation therapy have been reported to be different between right breast cancer and left breast cancer. Is this true in other reports as well?

We discussed about the different effects of radiotherapy in women with left- and right-sided breast cancers. However, this is based on the cancer location and its proximity to heart, not on cancer type. All studies confirmed that radiotherapy in left-sided cancer is more related with cardiac damage than  right-sided cancer. This is emphasized in the article.

#4 What measures, if any, are being taken to avoid the effects of radiotherapy on cardiac function (including the right heart)?

This is a very good question to which we do not have answer. All studies about cardiac protection were performed in patients treated with chemotherapy and influence of radiotherapy was neglected. We added this in the conclusion, as topic that deserves to be further investigated.

Reviewer 4 Report

Dear Authors,

This is an interesting review on a subject not sufficiently investigated since now of the effects of radiotherapy on right ventricular function of the heart. Authors collected  and analyzed the results of several studies on the effects of anticancer therapy on available echocardiographic and NMR indices of right ventricular function. This topic may also be important  due to the growing number and  the prolonged survival of patients after anticancer therapy. Some of them have impaired cardiac function, in some cases leading to advanced heart failure and can be evaluated for the treatment by left ventricular assist devices (LVAD). For these patients, the evaluation of the right ventricular function is of crucial importance.

Therefore, I agree with the conclusion about the importance of assessing right ventricular function in imaging of heart  of patients after anticancer therapy and it is an important argument for the publication of this review

Major Remarks

1.       As pointed out the main area of interest of the review was the deterioration of right cardiac function. However, there is a very strict interrelation between left and right ventricular function (left-right ventricle interdependence) and possible added effects of thickening  and stiffening of the pericardium that often to  an increased left and right end diastolic pressures. In advanced  cases those patients present a restrictive cardiomyopathy pattern.  . In my opinion, the aspects signaled above should be more stressed in the introduction to your article.

2.       Did you search the literature of the subject in a systematic way (for example, PubMed with key words or another method), if so please specify that in your text and which keywords you used.

Minor remarks

1.       An extensive language check should be performed, preferably by a native speaker.

2.       Line 105 Please use full name before the first appearance of acronym RIHD

3.       Table1 and 2 -please separate clearly each study ( in the main findings column it is difficult to follow which finding is the result of which study)

Author Response

We would like to thank you for the detailed review of our manuscript. We greatly appreciate the effort you made concerning your critique for the review of our study.

Major remarks:

  1. It is difficult to distinguish effect of radiotherapy on the LV from the RV because there is a significant interventricular dependence, as well as possible additional effects of thickening and stiffening of the pericardium that may increase RV and LV end-diastolic pressures. Different imaging parameters are used to overcome these issues with more or less success.
  2. The PubMed database was searched for all articles published in English up to October 31st 2022. Studies were identified by using terms: “right ventricle”, “cancer”, “radiotherapy”, “strain”, “systolic function”, “longitudinal strain” and “echocardiography”. This is added in the current version of manuscript.

Minor remarks:

  1. English language has been revised. Hopefully, it is better in this version of manuscript.
  2. RIHD has been explained, as you suggested.
  3. We increase spacing between different studies and we hope it is easier for understanding in this form.

We did our best to change manuscript according to your suggestions and provide all necessary explanations. We hope that the manuscript in its current form fulfills your and the Journal criteria for publication.